# The Effect of Chemotherapy Treatment on Cognitive Impairment and Clinical Symptoms in Hodgkin Lymphoma Patients

**DOI:** 10.3390/cancers17091488

**Published:** 2025-04-28

**Authors:** Dan Fayette, Veronika Juríčková, Iveta Fajnerová, Jiří Horáček, Tomáš Kozák

**Affiliations:** 1National Institute of Mental Health, Topolová 748, 25067 Klecany, Czech Republic; veronika.jurickova@nudz.cz (V.J.); iveta.fajnerova@nudz.cz (I.F.); jiri.horacek@nudz.cz (J.H.); 2Third Faculty of Medicine, Charles University, Ruská 87, Prague 10, 10100 Prague, Czech Republic; tomas.kozak@fnkv.cz; 3First Faculty of Medicine, Charles University, Kateřinská 1660/32, 12108 Prague, Czech Republic; 4Department of Clinical Hematology, University Hospital Královské Vinohrady, Šrobárova 50, 10034 Prague, Czech Republic

**Keywords:** Hodgkin lymphoma, chemotherapy, cognitive functions, cancer-related cognitive impairment, chemotherapy-related cognitive impairment, cancer

## Abstract

The cancer- or chemotherapy-related cognitive impairment is a common side effect occurring in patients with Hodgkin lymphoma. Our previous research revealed lower cognitive performance in patients with Hodgkin lymphoma compared to healthy controls. The current standard of care is having different specific treatment strategies for Hodgkin lymphoma patients with early-stage disease and those with advanced disease. The aim of our longitudinal study was to assess whether patients with more advanced disease and more aggressive treatment will have worse cognitive performance than early-stage patients treated with a milder form of chemotherapy. Our data do not suggest that patients with more advanced Hodgkin lymphoma or those undergoing a more intensive form of chemotherapy have greater cognitive impairment than patients with earlier-stage Hodgkin lymphoma treated with a milder form of chemotherapy. However, a limitation of our study is the smaller number of participants enrolled in the study.

## 1. Introduction

The cancer- or chemotherapy-related cognitive impairment (CRCI) is a common side effect occurring in patients with Hodgkin lymphoma (HL) [1,2], with an adverse effect on the quality of patient’s life, their work performance, social integration, and compliance with the treatment procedure.

One of the most important questions in this field is whether cognitive deficits arise during the cancer itself or during treatment. To this end, our previous study demonstrated that cognitive deficits in HL patients are already present before treatment [1]. However, several studies also reported the chemotherapy-related cognitive impairment [3,4,5,6]. These changes have been found to persist even in the immediate post-treatment period and even months or years after the treatment [2,3].

The current standard of care is having different specific treatment strategies for HL patients with early-stage disease and those with advanced disease. Patients with an early-stage disease are typically treated by combined modality strategies utilizing abbreviated courses of combination chemotherapy followed by involved-field radiation therapy (IF RT), e.g., ABVD (adriamycin, bleomycin, vinblastine, and dacarbazine) and 30 Gy IF RT, while those with advanced-stage disease receive a longer course of chemotherapy often without radiation therapy, e.g., BEACOPP (bleomycin, etoposide, adriamycin, cyclophosphamide, vincristine, procarbazine, and prednisone) [7]. The BEACOPP treatment regimen is a more effective tool in patients with advanced HL but is associated with greater toxicity than ABVD, whereas ABVD is better tolerated but less effective [8,9,10,11]. A more aggressive form of HL treatment is represented not only by more frequent exposure to doxorubicin, which is present in both ABVD and BEACOPP regimens, but also by more frequent exposure to cyclophosphamide, which is present only in the BEACOPP regimen. Cyclophosphamide (along with doxorubicin) reversibly reduces the number of mature neurons and synaptic transmission in the hippocampus, but unlike doxorubicin, it can also cross the blood–brain barrier [12].

To the best of our knowledge, no study has compared the influence of different types of treatment on the development of chemotherapy cognitive impairment in the longitudinal design. This approach is necessary not only to clarify the above-mentioned question of whether cognition is impaired by HL itself or only by subsequent treatment, but also to clarify the degree of neurotoxicity of specific treatment regimens. To fill this knowledge gap, we compared the cognitive performance before, immediately after, and 6 months after the treatment in HL patients with different types of chemotherapy treatment.

Given the above-mentioned differences between the treatment regimens, we expected greater toxicity for the treatment regimens using BEACOPP.

## 2. Methods

### 2.1. Participants

HL patients were recruited from the Department of Internal Medicine and Hematology of University Hospital Kralovske Vinohrady (FNKV) in Prague. HL patients of classical histological subtypes (lymphocyte predominance Hodgkin disease was excluded) were recruited prior to adjuvant chemotherapy and radiotherapy. Inclusion criteria were as follows: (a) classical HL diagnosed according to WHO classification, (b) age 18–70 years, (c) patients eligible for the treatment according to guidelines, and (d) signed informed-consent form. Exclusion criteria involved history of psychosis, substance abuse, mood disorder, organic mental disorder, serious head injury, or neurosurgery.

Patients at an early stage were treated with two cycles of ABVD and 30 Gy of IF RT. Patients at an intermediate stage were given two cycles of BEACOPP escalated (BEACOPPesc) plus two cycles of ABVD plus 30 Gy of IF RT. Patients at an advanced stage were treated with six cycles of BEACOPPesc. All the participants were recruited between April 2016 and December 2019. The study was approved by the local ethics committees of the NIMH and the FNKV. All respondents signed an informed-consent form to participate in the study and have their data processed for research purposes. Participation in the study was voluntary and without compensation.

### 2.2. Procedures

The neuropsychological examination took place at the National Institute of Mental Health (NIMH) in a separate quiet room and was provided by a trained psychologist (D.F., V.J., or I.F.). The assessment required approximately 90–120 min (including the breaks) to complete, and it always began at the same time in the morning. The data were collected from each participant three times: (1) before the treatment (a few days after establishing the diagnosis), (2) promptly after the treatment (approximately 6 months after the baseline assessment), and (3) 12 months after the baseline assessment. At the time of the examination and interpretation, the psychologists did not know which treatment group the person under examination belonged to. This information was known to the treating oncologist, and it was only released to the psychologists for statistical data analysis.

### 2.3. Cognitive Testing

The neurocognitive testing battery consisted of the Rey Auditory Verbal Learning Test (RAVLT); the Rey–Osterrieth Complex Figure Test (ROCFT); the Trail Making Test (TMT, parts A and B); the Verbal Fluency Test (VFT); the Logical Memory Subtest from the Wechsler Memory Scale—third edition, abbreviated (WMS-IIIa) [13]; the Continuous Performance Test 3 (CPT-3) [14]; four subtests of the Wechsler Adult Intelligence Scale—third edition (WAIS-III), i.e., digits span, similarities, digit symbols (coding), and letter-number sequencing [15]; and the Stroop Test [16]. For repeated measurements, alternative versions of AVLT, ROCFT (Taylor Figure in the second evaluation and the Medical College of Georgia Complex Figure I [17] in the third), and WMS-IIIa available for the Czech population were used. All 28 subtests were divided into 6 individual cognitive domains (attention/vigilance, memory and learning, working memory/flexibility, verbal memory and learning, speed of processing/psychomotor speed, and abstraction/executive functions). For more information on the division of individual tests into cognitive domains, see our previous studies [1,2].

We also included measures of levels of anxiety, depression, and experienced quality of life. This is because both increased anxiety and depression [18,19,20] and decrease in experienced quality of life are frequently found in cancer patients, including those with HL [21,22], and can affect cognitive performance [23,24]. All the participants therefore completed the self-reported Beck Depression Inventory (BDI-II) [25], and the Beck Anxiety Inventory (BAI-II) [26]. We used the Hamilton Depression Rating Scale (HAM-D) [27] and the Hamilton Anxiety Rating Scale (HAM-A) [28] for clinical assessment. Quality of life was measured by the Czech version of the World Health Organization Quality of Life (WHOQOL-BREF) [29].

### 2.4. Statistical Analyses

The data obtained in the cognitive assessments were transformed into z-scores (for more information see our previous studies [1,2]). The statistical analyses were processed using software JAMOVI 1.6.3. Descriptive statistics were used to analyze the frequencies, means, and standard deviations of the study variables. The raw scores of cognitive subtests were transformed into z-scores based on the normative data for the individual test methods. The normality of the score distribution was tested by the Shapiro–Wilk test. The data were not normally distributed among several variables, so nonparametric statistical methods were used for further analysis. The Friedman test and Wilcoxon signed-rank test (for post hoc tests) were used to evaluate the within-group differences. The between-group differences were analyzed by the non-parametric Mann–Whitney U test (for two groups) and Kruskal–Wallis test (for three groups). To indicate statistical significance, two-sided *p*-values of 0.05 were corrected using a flexible Bonferroni–Holm procedure for multiple comparison. Sociodemographic data were analyzed using Spearman’s correlation coefficient (age and education level) and Mann–Whitney U (sex).

## 3. Results

### 3.1. Description of the Total Sample

The final sample included 44 patients with HL (24 men; 54.5%) aged between 20 and 68 years of age (M = 36.94; SD = 11.88). The education level was distributed as follows: 7.1% (*n* = 3) of patients had an elementary education, 11.9% (*n* = 5) had a certificate of apprenticeship, 52.4% (*n* = 22) had a high school education, 26.2% (*n* = 11) had a university degree, and 2.4% (*n* = 1) had a post-doctoral education.

### 3.2. Division of the Total Sample into Two Groups and Their Comparison

The patients were divided into two groups according to the type of chemotherapy treatment received and its predicted toxicity to brain tissue. The ABVD regimen is generally considered to be less toxic than BEACOPPesc [12]. Therefore, we have included patients who were treated exclusively or mainly with this regimen in one group, while patients who were treated exclusively with the more aggressive form of BEACOPPesc have been included in the second group. Group one included all patients who were treated in regimens designed for the early and intermediate stages of disease, i.e., ABVD and 30 Gy of IF RT or two cycles of BEACOPPesc plus two cycles of ABVD plus 30 Gy of IF RT. The second group consisted of patients with advanced disease who were treated with six cycles of BEACOPPesc. A total of 23 people (52.3%) were included in the first group, and 21 people (47.7%) in the second group. The two groups did not differ in demographic variables. See Table 1 for more details.

Significant within-group differences were found in the mean scores of almost all cognitive domains, except for attention/vigilance (after Bonferroni–Holm correction) (Table 2). Surprisingly, in the period just after the end of chemotherapy (i.e., six months after the start of our follow-up), we observed only an improvement in cognitive performance in the whole group of patients; we did not observe a decrease in performance in any cognitive domain. Improvements in cognitive performance were observed in working memory/flexibility, speed of processing/psychomotor speed, and abstraction/executive functions. After recovery (i.e., one year after the start of follow-up), improvements to the first measurement were noted in visuospatial memory and learning, working memory/flexibility, verbal memory and learning, and speed of processing/psychomotor speed. Verbal memory and learning also improved significantly compared to the second measurement. Decreased performance was found only in abstraction/executive function, with lower scores in the third compared to the second measure.

We also monitored the changes in cognition over one year separately for both groups of HL patients. Cognitive performance improved or remained unchanged over time in most cognitive domains in both groups. A statistically significant decrease in performance was found only in abstraction/executive function in group 1, with lower scores in the third compared to the second measure. For more information on changes in cognitive performance over a one-year period in the two tested groups, see Table 3.

The results of the between-group comparisons showed no significant differences in cognitive performance between the two groups tested, either in the first or in any of the subsequent measures (Table 3).

Within-group differences showed significant differences in HAMD and BAI scores in both groups. However, after Bonferroni correction, only the subjective anxiety score remained significant, and only in group number two, i.e., the group of patients with more advanced stage Hodgkin lymphoma. Subjective perceptions of anxiety were significantly more pronounced in this group upon the first measurement than upon the two follow-up measurements. No differences in clinical assessment or subjective measurements of anxiety (HAMA and BAI-II) and depression (HAMD and BDI-II) were found between the two treatment groups either (Table 4).

In the next step, we investigated whether the results were dependent on the selected sociodemographic data. Our data showed no association between sex, educational attainment, or age and depth of cognitive impairment. Further information on the association between these sociodemographic and cognitive test scores can be found in Appendix A. Post hoc, we divided the total sample according to the treatment protocol into three groups (for early stage, intermediate stage, and advanced stage). We report the results of this post hoc comparison in Appendix B.

## 4. Discussion

This is the first study to evaluate the effect of different types of HL treatment on cognitive function in patients in a longitudinal design. Our study is also the first to compare ABVD (for early stages of HL) and BEACOPPesc (for advanced stages of HL) regimens. Our previous results showed that the cognitive deficits are already present in HL patients prior to treatment. Although the cognitive performance of HL patients tends to improve or remain stable over time (as shown in this study), compared to the control group, patients’ cognitive performance lags in some domains [2]. We suggest that the improvement in patients’ cognitive performance may be due to a learning effect when using the same or similar test methods, even though we tried to avoid this effect by using alternative versions of the tests and measuring after a six-month break. Given that our previous study has demonstrated cognitive impairment in patients, we felt it was important to investigate whether disease stage itself also influences the depth of CRCI. Based on our data, we were unable to demonstrate an association between disease progression or cognitive decline in any of the cognitive domains examined. Our data also failed to demonstrate a difference between treatment protocols in terms of their effect on cognitive function in the patients tested.

Our negative findings are not consistent with some previous studies, which reported that some cognitive functions deteriorate depending on the stage of the disease and the number of cycles of chemotherapy [30,31]. The metabolism of the prefrontal cortex, cerebellum, medial cortex, and limbic regions of the brain was negatively correlated with the number of cycles of chemotherapy [30]. The study of Magyari [31] has found that a more pronounced impairment of psychomotor tempo is present in patients at a more advanced stage of the disease. The authors attribute this finding to the number of cycles of chemotherapy. Our study has not established that the stage of the disease or length of treatment are related to the severity of cognitive impairment, as no differences have been found between the early and advanced stages of HL or patients with different treatment protocols. These contradictory findings may be due to different methods used to evaluate cognitive performance (the above-mentioned authors used neuroimaging or different neuropsychological methods than were applied in our research). However, our study provides a thorough assessment of cognitive performance across all cognitive domains. In addition, previous studies compared different types of treatment. In contrast, the Trachtenberg study, which compared the same treatment regimens as our study (i.e., ABVD vs. BEACOPPesc), reached the same conclusion. However, this study was based (like our study) on a small number of respondents (*n* = 50). Thus, the question remains whether the lack of difference between the two patient’s cohorts is due to the treatment itself or to the smaller sample of respondents.

Our observation that patients in the advanced stage of the disease (i.e., those who were subsequently treated with 6 × BEACOPP) showed the highest subjectively perceived levels of anxiety before treatment compared to subjectively perceived anxiety upon follow-up measures. However, this higher perceived level of anxiety was not statistically significantly higher than in patients at earlier stages of the disease and did not affect cognitive test scores overall. On average, patients in advanced stages of the disease performed on par with patients in less advanced stages of HL. The relationship between psychological factors and patients’ cognitive performance was addressed in detail in our previous report, which did not identify significant effects of emotional factors on cognitive performance in patients with HL [2]. We found no significant differences in cognitive performance based on education, gender, and age.

We administered a valid and standardized neuropsychological battery; however, some cognitive domains were not satisfactorily covered with at least two tests. Even if we used some alternative versions of the tests, we could not exclude the ceiling effect. We also admit that the psychometric properties of these versions have not been adequately explored. The significant limitation of our study is a small number of respondents. We acknowledge that the small number of respondents in our study may have influenced the results obtained, limiting the strength of our conclusions. Studies with small numbers of respondents may be subject to type 2 errors: they may not always be able to adequately capture differences between groups. Therefore, it would be advisable to repeat the measurement on a larger sample. On the other hand, our sample groups were well aligned with demographic data and were also balanced in terms of the number of respondents. Another merit of our research is that it employed a longitudinal design study.

Our findings suggest that intensive chemotherapy protocols do not worsen cognitive functions, thus alleviating concerns for both patients and physicians. Future studies should validate these results using a larger cohort.

## 5. Conclusions

Our study did not show that disease stage or treatment protocol affected the depth of cognitive impairment in HL patients. Our data also did not show that intensive forms of chemotherapy treatment for HL pose a greater risk to brain health than milder forms of anti-cancer treatment.

## Figures and Tables

**Table 1 cancers-17-01488-t001:** Comparison of groups in terms of sociodemographic data.

		Group 1	Group 2	Group Difference
HL stage	Early	13	0	-
	Intermediate	10	0	-
	Advanced	0	21	-
Age		M = 37.5; SD = 14.0; 21–67	M = 36.3; SD = 9.27; 19–54	*p* = 0.798
Sex	Male	12	12	*p* = 0.075
	Female	11	9	
Education	Elementary school	0 (0%)	3 (15.8%)	*p* = 0.798
	Certificate of apprenticeship	2 (8.7%)	3 (15.8%)	
	High school education	13 (56.5%)	9 (47.4%)	
	University degree	7 (30.4%)	4 (21.1%)	
	Post-doctoral education	1 (4.3%)	0 (0%)	

**Table 2 cancers-17-01488-t002:** Results of repeated cognitive assessment in all HL patients (*n* = 44) and health controls expressed in cognitive domain z-scores.

Cognitive Domain	Group	1st Visit M (SD)	2nd Visit M (SD)	3rd Visit M (SD)	Friedman Test	Within-Group Differences (Significant)
Attention/vigilance	Patients	0.254 (0.878)	0.275 (0.281)	0.576 (1.27)	X^2^ = 4.47, *p* = 0.107	1–3 *^#^
	Controls	0.32 (0.41)	0.68 (1.23)	0.44 (0.34)	X^2^ = 8.77, *p* = 0.012	1–3 *^#^
Visuospatial memory and learning	Patients	0.215 (0.823)	0.607 (1.18)	0.899 (1.61)	X^2^ = 15.1, *p* ≤ 0.001	1–3 *** 2–3 **
	Controls	0.36 (1.01)	1.16 (1.08)	1.44 (1.15)	X^2^ = 40.26, *p* < 0.001	1–2 ***, 1–3 ***, 2–3 *^#^
Working memory/flexibility	Patients	−0.295 (0.586)	−0.043 (0.623)	0.016 (0.678)	X^2^ = 15.0, *p* ≤ 0.001	1–2 ** 1–3 ***
	Controls	0.00 (0.78)	0.20 (0.74)	0.15 (0.84)	X^2^ = 5.57, *p* = 0.062	1–2 **
Verbal memory and learning	Patients	−0.372 (0.882)	−0.209 (0.654)	0.026 (0.560)	X^2^ = 13.4, *p* = 0.001	1–3 *** 2–3 **
	Controls	0.36 (0.59)	0.24 (0.73)	0.55 (0.68)	X^2^ = 5.02, *p* = 0.081	1–3 **, 2–3 **
Speed of processing/psychomotor speed	Patients	−0.467 (0.599)	−0.252 (0.650)	−0.122 (0.840)	X^2^ = 21.6, *p* ≤ 0.001	1–2 ***, 1–3 *** 2–3 *^#^
	Controls	−0.02 (0.63)	0.16 (0.62)	0.28 (0.71)	X^2^ = 10.71, *p* = 0.005	1–2 *,1–3 ***, 2–3 **
Abstraction/executive functions	Patients	−0.472 (0.726)	−0.231 (0.708)	−0.434 (0.769)	X^2^ = 11.9, *p* = 0.003	1–2 *** 2–3 **
	Controls	0.55 (0.72)	0.22 (0.97)	0.47 (0.74)	X^2^ = 4.76, *p* = 0.092	

Notes: Levels of significance are * *p* < 0.05, ** *p* < 0.01, and *** *p* < 0.001. ^#^ Did not survive the Bonferroni–Holm correction (level set at *p* < 0.017).

**Table 3 cancers-17-01488-t003:** Cognitive performance results from treatment protocol affiliation (two groups) and results of the within-group differences and between-group differences in cognitive domains expressed in z-scores.

Cognitive Domain	Treatment	1st Visit M (SD)	2nd Visit M (SD)	3rd Visit M (SD)	Friedman Test	Within-Group Differences (Significant)	Between-Group Differences Mann–Whitney U Test
Attention/vigilance	T1	0.372 (1.14)	0.301 (0.270)	0.328 (0.253)	X^2^ = 3.33, *p* = 0.189		1. v: U = 194.0 *p* = 0.450 2. v: U = 170.0 *p* = 0.583 3. v: U = 109.0 *p* = 0.884
	T2	0.123 (0.445)	0.249 (0.297)	0.825 (1.77)	X^2^ = 4.47, *p* = 0.107		
Visuospatial memory and learning	T1	0.389 (0.919)	0.242 (1.20)	1.19 (1.27)	X^2^ = 13.7, *p* = 0.001	1–3 *** 2–3 ***	1. v: U = 131.0 *p* = 0.161 2. v: U = 124.0 *p* = 0.062 3. v: U = 117.0 *p* = 0.502
	T2	0.00 (0.648)	0.991 (1.06)	0.627 (1.87)	X^2^ = 7.75, *p* = 0.021	1–2 ** 1–3 **	
Working memory/flexibility	T1	−0.287 (0.662)	−0.070 (0.704)	0.086 (0.670)	X^2^ = 7.0, *p* = 0.030	1–3 **	1. v: U = 160.0 *p* = 0.601 2. v: U = 169.0 *p* = 0.748 3. v: U = 95.5 *p* = 0.694
	T2	−0.304 (0.495)	−0.013 (0.537)	−0.049 (0.703)	X^2^ = 8.40, *p* = 0.015	1–2 **^#^ 1–3 **	
Verbal memory and learning	T1	−0.238 (0.934)	−0.154 (0.732)	−0.002 (0.576)	X^2^ = 6.00, *p* = 0.050	1–3 **	1. v: U = 148.0 *p* = 0.257 2. v: U = 152.0 *p* = 0.574 3. v: U = 90.0 *p* = 0.981
	T2	−0.528 (0.815)	−0.266 (0.575)	0.052 (0.565)	X^2^ = 7.43, *p* = 0.024	1–3 ** 2–3 *^#^	
Speed of processing/psychomotor speed	T1	−0.550 (0.629)	−0.344 (0.701)	−0.164 (0.859)	X^2^ = 12.0, *p* = 0.002	1–2 ** 1–3 ***	1. v: U = 138.0. *p* = 0.243 2. v: U = 150.0 *p =* 0.393 3. v: U = 99.0 *p =* 0.813
	T2	−0.364 (0.561)	−0.150 (0.591)	−0.083 (0.850)	X^2^ = 9.73, *p* = 0.008	1–2 *^#^ 1–3 ***	
Abstraction/executive functions	T1	−0.381 (0.617)	−0.041 (0.742)	−0.388 (0.689)	X^2^ = 9.68, *p* = 0.008	1–2 ** 2–3 **	1. v: U = 164.0 *p =* 0.223 2. v: U = 124.0 *p =* 0.097 3. v: U = 113.0 *p =* 0.338
	T2	−0.567 (0.831)	−0.421 (0.637)	−0.500 (0.894)	X^2^ = 3.45, *p* = 0.178		

Notes: Levels of significance are * *p* < 0.05, ** *p* < 0.01, and *** *p* < 0.001. ^#^ Did not survive the Bonferroni–Holm correction (level set at *p* < 0.017).

**Table 4 cancers-17-01488-t004:** Descriptive statistics and statistical comparisons in the subjective measures and clinical assessment of anxiety and depression symptoms in a cohort divided into two groups.

Variables	Visit	Group 1 Mean (SD)	Group 2 Mean (SD)	Between-Group Differences Mann–Whitney U Test	Within-Groups: Friedman	Significance Within Groups
HAMD	1st	6.70 (5.01)	6.71 (5.47)	U = 241.0; *p* = 0.999	G1: X^2^ = 7.92, *p* = 0.019 G2: X^2^ = 6.33, *p* = 0.042	G1: 1–2 *^#^ 1–3 ** G2: 1–2 *^#^ 1–3 **^#^
	2nd	2.86 (2.66)	3.60 (4.03)	U = 211.0; *p* = 0.828		
	3rd	3.50 (3.71)	3.27 (3.59)	U = 142.0; *p* = 0.787		
HAMA	1st	6.83 (4.31)	7.38 (5.20)	U = 235.0; *p* = 0.888	G1: X^2^ = 5.47, *p* = 0.065 G2: X^2^ = 4.93, *p* = 0.085	
	2nd	3.68 (3.72)	5.00 (3.67)	U = 166.0; *p* = 0.170		
	3rd	4.80 (5.80)	4.20 (4.39)	U = 147.0; *p* = 0.933		
BAI	1st	6.70 (4.79)	8.52 (6.10)	U = 205.0; *p* = 0.396	G1: X^2^ = 8.13, *p* = 0.017 G2: X^2^ = 12.6, *p* = 0.002	G1: 1–2 *^#^ 1–3 ** G2: 1–2 * 1–3 ***
	2nd	4.14 (3.91)	7.16 (5.79)	U = 137.0; *p* = 0.058		
	3rd	3.85 (3.18)	4.53 (5.07)	U = 140.0; *p* = 0.749		
BDI	1st	2.87 (2.36)	4.05 (4.03)	U = 207.0; *p* = 0.413	G1: X^2^ = 1.03, *p* = 0.597 G2: X^2^ = 0.154, *p* = 0.926	
	2nd	2.36 (2.11)	3.75 (3.93)	U = 182.0; *p* = 0.336		
	3rd	1.72 (1.41)	2.93 (2.60)	U = 104.0; *p* = 0.254		

Notes. G1—group 1; G2—group 2. Level of significance: * *p* < 0.05, ** *p* < 0.01, and *** *p* < 0.001. ^#^ Did not survive the Bonferroni–Holm correction (level set at *p* < 0.017).

## Data Availability

The data presented are available upon request to the corresponding author.

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
