# Peer review of "The Effect of Chemotherapy Treatment on Cognitive Impairment and Clinical Symptoms in Hodgkin Lymphoma Patients"

_cancers, 2025, doi:10.3390/cancers17091488_

Round 1

Reviewer 1 Report

Comments and Suggestions for Authors

The topic is very interesting and the paper is well written.

some minor revisions are necessary in my opinion.

1) participants section: could the authors clearly specify inclusion and exclusion criteria?

2) line 129: HAM-D and HAM-A should be considered clinician-based rather than "objective evaluation" tools.

3) line 240-241: neuroimaging data are considered as a method to evaluate cognitive performance and other neuropsychological methods are not specified. could the authors specify this sentence?

4) Was the diffusion of illness to the meninges present in any of the patients?

5) discussion section (line 264 - 266) and conclusion section: I think that the sample number is small and does not allow to draw so clear conclusions. More caution is necessary in the interpretation of the results.

Author Response

Manuscript ID: cancers-3564202

Response to Reviewer 1

We appreciate the time and effort that you have dedicated to providing your valuable feedback on our manuscript. We are grateful for the insightful comments. We hope that the revised manuscript incorporates sufficient changes addressing all your comments. A point-by-point response to your comments and concerns is provided below.

Reviewer: The topic is very interesting, and the paper is well written. Some minor revisions are necessary in my opinion.

Authors: Thank you for your appreciation. We have done the revisions as you have suggested.

Comment 1: participants section: could the authors clearly specify inclusion and exclusion criteria?

Response 1: Thank you for your comment. We agree that the inclusion and exclusion criteria are important information that we originally neglected to add. In line with this request, we have included the requested information. The required modification can be found on page 3, lines 104-110.

“HL patients of classical histological subtypes (lymphocyte predominance Hodgkin disease was excluded) were recruited prior to adjuvant chemotherapy and radiotherapy. Inclusion criteria were as follows: a) classical HL diagnosed according to WHO classification, b) age 18-70 years; c) patients eligible for the treatment according to guidelines d) signed Informed consent. Exclusion criteria involved: history of psychosis, substance abuse, mood disorders, organic mental disorder, serious head injury or neurosurgery. “

Comment 2: line 129: HAM-D and HAM-A should be considered clinician-based rather than "objective evaluation" tools.

Response 2: Thank you for your comment, we agree with the comment. We originally called the HAM-A and HAM-D questionnaires objective because we saw them as a kind of counterbalance to the self-assessment (subjective scales). However, we recognize that this is a misleading formulation, so we have followed your recommendation and modified the wording. The required modification can be found on page 4, lines 150-151.

“We used the Hamilton Depression Rating Scale (HAM-D) [27] and the Hamilton Anxiety Rating Scale (HAM-A) [28] for clinical assessment “.

Comment 3: line 240-241: neuroimaging data are considered as a method to evaluate     cognitive performance and other neuropsychological methods are not specified. could the authors specify this sentence?

Response 3: Thank you for pointing this out. The sentence was meant to imply that methodological choices will always affect the results, i.e. if the authors used brain imaging methods, while we assessed cognition through performance on psychological tests, this may explain the differences in the study results. Similarly, the choice of psychological tests used as outcome measure in various studies may also influence the obtained findings. We have modified the wording in the revised text to make the message clearer.  The required modification can be found on page 9, lines 275-277.

“These contradictory findings may be due to different methods used to evaluate cognitive performance (the above authors used neuroimaging or different neuropsychological methods than were applied in our research) “.

Comment 4: Was the diffusion of illness to the meninges present in any of the patients?

Response 4: No, in HL almost never involves CNS (unlike NHL) that´s why CNS involvement exclusion is not a standard part of staging work up in HL.

Comment 5: discussion section (line 264 - 266) and conclusion section: I think that the                      sample number is small and does not allow to draw so clear conclusions. More caution is necessary in the interpretation of the results.

Response 5: Thank you for your comment. We are aware that our sample contains a small number of respondents, which may have affected the results of our study. Following your recommendation, we have modified the wording of our results interpretation to make it clear that our findings show some tendency that should be confirmed (or refuted) by future studies with a larger sample. The required modification can be found on page 10, lines 303-307.

“We acknowledge that the small number of respondents in our study may have influenced the obtained results, limiting the strength of our conclusions. Studies with small numbers of respondents may be subject to type 2 error, i.e. they may not always be able to adequately capture differences between groups. Therefore, it would be advisable to repeat the measurement on a larger sample “.

Reviewer 2 Report

Comments and Suggestions for Authors

The aim of the study was to assess whether a more intensive type of chemotherapy causes greater cognitive impairment in patients with Hodgkin lymphoma. In the current study, no significant differences were found between the groups treated with ABVD + RT versus escalated BEACOPP. The strength of the study is that the cognitive tests were performed at 3 time points: 1) before the treatment (a few days after establishing the diagnosis); 2) promptly after the treatment (approximately 6 months after the baseline assessment), and 3) 12 months after the baseline assessment. The weakness of the study is the small cohort of 44 patients.

In current study, no significant difference was identified between patients with early stage and  advanced-stage HL patients in terms of cognitive impairment. These findings are comparable to those reported in a group of 50 HL survivors by Trachtenberg et al (ref 4).

While the topic is important and currently several studies are ongoing, it is yet possible that the cohort in the presented study was too small to draw definitive conclusions. This should be thoroughly  elaborated in the Discussion.

It would be also interesting to compare the findings between males and females and according to patients’ age.

Another issue worth considering is whether patients with lower IQ do have a higher degree of cognitive impairment than those who initiate treatment with higher cognitive reserve. Since the study population also had a baseline measurement, grouping according to the initial score and following the changes would be valuable.

Minor points:

The abbreviation “IF RRT” appearing in several places in the text needs to be expanded.  Did the authors actually mean “IFRT” (involved-field radiation therapy)?

Table 1: Please add the values obtained in the control normal population.

Table 2: It might be useful to compare the T1 and T2 group.  I might have missed this, but have not found a comparison of the executive function between the groups.

Please consider adding a table comparing the demographics (age, gender, HL stage, B symptoms and cumulative drugs exposure) between group 1 and group 2.

Author Response

Manuscript ID: cancers-3564202

Response to Reviewer 2

We appreciate the time and effort that you have dedicated to providing your valuable feedback on our manuscript. We are grateful for the insightful comments. We hope that the revised manuscript incorporates sufficient changes addressing all your comments. A point-by-point response to your comments and concerns is provided below.

Reviewer: The aim of the study was to assess whether a more intensive type of chemotherapy causes greater cognitive impairment in patients with Hodgkin lymphoma. In the current study, no significant differences were found between the groups treated with ABVD + RT versus escalated BEACOPP. The strength of the study is that the cognitive tests were performed at 3 time points: 1) before the treatment (a few days after establishing the diagnosis); 2) promptly after the treatment (approximately 6 months after the baseline assessment), and 3) 12 months after the baseline assessment. The weakness of the study is the small cohort of 44 patients.

Authors: Thank you for your appreciation. We agree that our study contains a low number of respondents.

Reviewer: In current study, no significant difference was identified between patients with early stage and advanced-stage HL patients in terms of cognitive impairment. These findings are comparable to those reported in a group of 50 HL survivors by Trachtenberg et al (ref 4).

Authors: Thank you for bringing this study to our attention. We have included the results of this study in the revised text. The modification can be found on page 9, lines 281-286.

“In contrast, the Trachtenberg study, which compared the same treatment regimens as our study (i.e., ABVD vs BEACOPPesc), reached the same conclusion. However, this study was based (like our study) on a small number of respondents (n = 50). Thus, the question remains whether the lack of difference between the two patient’s cohorts is due to the treatment itself or to the smaller sample of respondents“.

Reviewer: While the topic is important and currently several studies are ongoing, it is yet possible that the cohort in the presented study was too small to draw definitive conclusions. This should be thoroughly elaborated in the Discussion.

Authors: We agree with the comment. We have elaborated more on this information in the discussion. The modification can be found on page 10, lines 306-310. We have also modified the wording throughout the text so that the interpretation of our results is not exaggerated.

“We acknowledge that the small number of respondents in our study may have influenced the obtained results, limiting the strength of our conclusions. Studies with small numbers of respondents may be subject to type 2 error, i.e. they may not always be able to adequately capture differences between groups. Therefore, it would be advisable to repeat the measurement on a larger number of respondents “

Reviewer: It would be also interesting to compare the findings between males and females and according to patients’ age.

Authors: Thank you for your suggestion. We have performed additional analysis according to your suggestions, by age and gender. No significant differences in the depth of cognitive impairment were found to indicate an effect of age or gender. We have added the information to the text on page 8, line 238-241 and on page 9, lines 296-297. A table with more detailed information has been added to Appendix A (Table A2 and A3, p. 12-13).

Similar results were obtained when the respondents were divided into two groups according to treatment and when the whole population was analyzed. In the text, we present only the results divided by group to maintain the theme of the paper, i.e., the comparison of two differently treated groups of patients.

Reviewer: Another issue worth considering is whether patients with lower IQ do have a higher degree of cognitive impairment than those who initiate treatment with higher cognitive reserve. Since the study population also had a baseline measurement, grouping according to the initial score and following the changes would be valuable.

Authors: Thank you for your suggestion. We could not do analyses based on IQ because we did not measure it initially. Also, we could not consider the results of the first measurements as a suitable cut-off criterion because our previous analyses showed that cognitive impairment is already present before chemotherapy (Juríckova, 2023). Thus, we found it most useful as a criterion to look at how performance is affected by educational attainment (as an estimate of premorbid intelligence). We were unable to find any correlation between educational attainment and test scores on the individual measures. We have added this information on page 8, lines 238-241 and on page 9, lines 296-297. We have placed the table in Appendix A (Table A1, p. 11).

Ref: Juríčková, V.; Fayette, D.; Jonáš, J.; Fajnerová, I.; Kozák, T.; Horáček, J. Pretreatment Cancer-Related Cognitive Impairment in Hodgkin Lymphoma Patients. Curr. Oncol. 2023, 30(10), 9028-9038. https://doi.org/10.3390/curroncol30100652

Minor points:

Comment 1: The abbreviation “IF RRT” appearing in several places in the text needs to be expanded.  Did the authors actually mean “IFRT” (involved-field radiation therapy)?

Response 1: Thank you for your comment. Yes, IF RRT is the same IF RT (involved field radiotherapy). We have corrected the abbreviation in the text.

Comment 2: Table 1: Please add the values obtained in the control normal population.

Response 2: We added the results obtained by the control group to Table 1 (now Table 2).

Comment 3: Table 2: It might be useful to compare the T1 and T2 group.  I might have missed this, but have not found a comparison of the executive function between the groups.

Response 3: We are not sure we understand the question well. However, a comparison of executive functions between the two groups is provided in Table 2 (now Table 3) in the last row (page 7). No differences were found in the between-group comparison.

Comment 4: Please consider adding a table comparing the demographics (age, gender, HL stage, B symptoms and cumulative drugs exposure) between group 1 and group 2.

Response 4:

Thank you for this important comment. In the revised manuscript, we have included a proposed demographic table 1 comparing age, sex, and HL stage between group 1 and group 2 on page 5. However, we could not provide information on B symptoms and cumulative drug exposure because we do not have it. Calculating cumulative drug exposure at the level of each drug used would be too complex and beyond the scope of the current analysis. We thank you for your understanding.

Reviewer 3 Report

Comments and Suggestions for Authors

This is a well formulated research study investigating a very worthwhile question: does intensity of treatment for HL impact cognitive impairment.  The results, while they conflict with some other studies, appear to have high validity.  The researchers chose a very well validated battery of neuropsychological tests, and did what they could to account for any learning effects from multiple episodes of test taking.  One question I have is whether educational level had anything to do with cognitive changes over time.  Their sample appeared to have fairly few individuals with a college education or higher.  

Author Response

Manuscript ID: cancers-3564202

Response to Reviewer 3

We appreciate the time and effort that you have dedicated to providing your valuable feedback on our manuscript. We are grateful for the insightful comments. We hope that the revised manuscript incorporates sufficient changes addressing all your comments.

Reviewer: This is a well formulated research study investigating a very worthwhile question: does intensity of treatment for HL impact cognitive impairment.  The results, while they conflict with some other studies, appear to have high validity.  The researchers chose a very well validated battery of neuropsychological tests, and did what they could to account for any learning effects from multiple episodes of test taking.  One question I have is whether educational level had anything to do with cognitive changes over time.  Their sample appeared to have fairly few individuals with a college education or higher.  

Authors: Thank you for your appreciation. We tried to correlate test scores on each measurement with educational attainment. We were unable to find any correlation between educational attainment and test scores on the individual measures. We have added this information to the text on page 8, lines 234-240 and on page 10, lines 295-299. We have placed the table in Appendix A (Table A1, p. 11).